# Dual-Process Theory of Thought and Inhibitory Control: An ALE Meta-Analysis

**DOI:** 10.3390/brainsci14010101

**Published:** 2024-01-20

**Authors:** Giorgio Gronchi, Gioele Gavazzi, Maria Pia Viggiano, Fabio Giovannelli

**Affiliations:** Department of Neuroscience, Psychology, Drug Research and Child’s Health (NEUROFARBA), University of Florence, 50135 Florence, Italy; giorgio.gronchi@unifi.it (G.G.); gioele.gavazzi@unifi.it (G.G.); fabio.giovannelli@unifi.it (F.G.)

**Keywords:** dual process, meta-analysis, left inferior frontal gyrus, insula, pre-SMA, DMN, PARCS theory

## Abstract

The dual-process theory of thought rests on the co-existence of two different thinking modalities: a quick, automatic, and associative process opposed to a slow, thoughtful, and deliberative process. The increasing interest in determining the neural foundation of the dual-process distinction has yielded mixed results, also given the difficulty of applying the fMRI standard approach to tasks usually employed in the cognitive literature. We report an activation likelihood estimation (ALE) meta-analysis to investigate the neural foundation of the dual-process theory of thought. Eligible studies allowed for the identification of cerebral areas associated with dual-process theory-based tasks without differentiating between fast and slow thinking. The ALE algorithm converged on the medial frontal cortex, superior frontal cortex, anterior cingulate cortex, insula, and left inferior frontal gyrus. These structures partially overlap with the cerebral areas recurrently reported in the literature about the neural basis of the dual-process distinction, where the PARCS theory-based interpretation emphasizes the role of the right inferior gyrus. The results confirm the potential (but still almost unexplored) common ground between the dual-process literature and the cognitive control literature.

## 1. Introduction

Within cognitive psychology, a highly shared view is that human reasoning and decision making depend on two distinct systems [1,2]. In the last 50 years, experimental psychology has empirically investigated this duality in deductive and inductive reasoning, decision making, and social judgement. The basic idea is that our responses are determined by two conflicting processes, where the first is often described as fast, effortless, automatic, and associative-based, whereas the second is slow, effortful, controlled, and rule-based. Some authors have gone further into developing general theories about mental architectures based on this distinction. According to these theories, human beings actually have two minds. These systems have been called in several ways, depending on the authors (fast thinking, associative, or System 1 vs. slow thinking, deliberative, or System 2). As Frankish and Evans wrote, “These all-encompassing theories are sometimes referred to as dual-system theories, in contrast to more localized dual-process ones, but ‘dual process theory’ is also used as an umbrella term for both” ([1], p. 1).

Nowadays, the dual-process framework has become a hallmark of the investigation of human cognition, enjoying an ever-growing popularity in psychological research [3]. As a matter of fact, Sherman et al. [4] wrote that “the emergence of the dual-process theories has been one of the most significant developments in the history of scientific psychology” ([4], p. xi). A great amount of cognitive and decision-making research takes place within this framework [5], with an extensive and dedicated work being aimed to clarify how the two systems interact [6], including the debate about the necessity to include a third system in the framework [7]. Psychometric research has focused on the development of a measure of propensity to employ these two styles of thinking [8,9,10]. More generally, the dual-process framework has influenced a variety of fields, such as moral psychology [11,12], social psychology [4,13], behavioral change and choice architectures [14,15], numerical cognition [16], understanding how fake news propagate [17,18], religious and paranormal beliefs [19,20,21,22], and even the scientific investigation of illusionist tricks [23]. Taking into account the variety of specific theories and terminology, as well as the different fields of investigation, in this paper, we will use the expressions fast and slow thinking to distinguish the two forms of processing [24].

Given the importance of the dual-process theory framework, a great interest in investigating the neural basis of the dual-process distinction has emerged. Exploiting brain imaging techniques, in recent years, several studies have been published with the aim of understanding cortical areas associated with the two forms of reasoning. Despite the increasing number of papers devoted to this issue, the overall picture remains fragmented due to the variety of topics investigated, methodology, and approaches [25]. In the following, after a brief description of the dual-process theory of thought, we will report an ALE meta-analysis gathering data from studies which employed fMRI to investigate the neural basis of the two systems of thinking.

### 1.1. The Dual-Process Theory of Thought

The basic idea that cognitive processes may be subdivided into the two broad categories of slow and fast thinking is supported by extensive empirical evidence. A task commonly employed to investigate the dual-process distinction is related to the so-called belief bias (i.e., judging a believable conclusion as more acceptable than an unbelievable conclusion regardless of the logical validity of the argument). For example, Evans et al. [26] presented categorical syllogism (e.g., Premise 1: All birds have wings; Premise 2: All crows are birds; Conclusion: Therefore, all crows have wings), manipulating the logical validity of the syllogism and the believability of the conclusion. When asking about the level of acceptability of the conclusion, the authors observed that the participants’ responses were determined by the interaction of both believability and the logical validity of the argument. Decades of research [27,28,29] confirmed the interpretation that slow thinking is responsible for the evaluation of the logical structure of the argument regardless of content, whereas fast thinking automatically forms the sensation that believable conclusions are acceptable (e.g., all crows have wings) and unbelievable conclusions are not (e.g., all apples are meat products); see [27] for a discussion. In the case of incompatibility between believability and logical validity, a conflict between fast and slow thinking emerges. The nature of this conflict and the more general questions related to how the two systems interact are at the core of the current debate in the dual-process theory literature. According to serial models (namely, the default–interventionist model [30,31]) fast thinking generates responses by default. Then, slow thinking may be activated and potentially intervene, assuming the availability of adequate resources. On the contrary, parallel models [32,33] postulate that fast and slow thinking occur simultaneously with continuous monitoring. A third option is represented by the “hybrid two-stage model”, which splits the function of slow thinking in two distinct processes: an always active, shallow, and analytics-based monitoring process and an optional, deeper, slow process. The former detects potential conflicts, with fast thinking activating the latter in case of necessity [30,34,35].

Taking into account the differences among these models, a pretty common feature is the inhibitory role carried out by slow thinking (or an equivalent counterpart; see [36] for a discussion) with respect to the fast-thinking response. This crucial aspect emerges clearly in the parallels between dual-process research and inhibitory control executive functioning research [36]. The former deals with high-level reasoning tasks with response times of tens of seconds, whereas the cognitive control literature deals with reaction times in the range of milliseconds; however, in both cases, a dominant automatic response must be inhibited to complete the requested task. Several examples can be found in the executive functioning literature (Stroop task, stop-signal task, go–no go [37,38,39]), as well as in the dual-process framework (the belief bias, the Linda problem, the cognitive reflection test [8]). The cognitive reflection test (CRT) is a gold standard in the psychometric approach to the dual-process distinction. Its original form comprises three reasoning questions, where each one has a (normatively wrong) obvious intuitive answer and a less accessible (but normatively correct) answer that requires some analytical deliberation. The general view is that slow thinking (i.e., an effortful and analytical modality) attempts to inhibit the apparently natural response (induced by effortless fast thinking).

### 1.2. Neural Correlates of Fast and Slow Thinking

Research on the neural correlates of high-level reasoning and decision-making tasks is comparatively less frequent than studies about shorter-time-course tasks related to basic cognitive processes [40]. Nevertheless, in recent years, an increasing number of researchers have challenged the methodological issues which arise when investigating the neural activity underpinning fast and slow thinking.

The neural basis of dual-process theory has been investigated by means of electrophysiology (see [40] for a review) and the functional magnetic resonance imaging (fMRI) technique. Focusing on the latter, most studies have found patterns of activation of the prefrontal cortex, particularly in regions involved in cognitive control processes [36,41,42,43]. A shared view is summarized by Evans and Stanovich [31] as follows: “conflict detection is indicated by activation of the anterior cingulate cortex and the override of belief-based responding with reasoning signaled by activation of the regions of the right prefrontal cortex known to be associated with executive control.” ([31], p. 233).

However, the specific structures associated with the two systems are still controversial. For example, De Neys et al. [44] found that problems characterized by a conflict between the two systems were associated with greater activation of the anterior cingulate cortex (ACC) and, in case of a positive resolution, with the additional activation of the right inferior frontal gyrus (r-IFG). These structures are commonly associated with conflict detection and response inhibition, respectively [45,46,47,48]. On the contrary, Vartanian et al. [49] found that the resolution of CRT scores predicted activation of the posterior cingulate cortex (PCC) with the same problems inducing a conflict between the two systems.

The PCC is a cerebral structure associated with the default-mode network (DMN), the role of which within the dual-process theory of thought has been proposed by several authors [25,49,50]. Besides the PCC, the DMN includes the medial prefrontal cortex, the inferior parietal lobule, the lateral temporal cortex, the dorsal medial prefrontal cortex, and the hippocampal formation. This neural system is active during passive rest and mind wandering, and it is thought to have a role in different functions, including thinking about others and the self, remembering the past, and thinking about the future. DMN activation at rest in the absence of external stimulation or a task to perform is believed to act as an autopilot, as opposed to the anterior insula, which is associated with awareness and task-related attention [51,52]. For this reason, the DMN has been hypothesized to be a possible neural foundation of fast thinking, which could act as a sort of automatic decision-making mode [25,49,50].

With regard to the right prefrontal cortex, some studies emphasized the role of the r-IFG [53,54,55], an area that plays a crucial part in the reactive circuit that responds to novel and salient stimuli [51,52]. From this perspective, the r-IFG should override the System 1 response. This hypothetical role of the r-IFG has induced researchers to find a correlation between the dual-process theory of thought and the PARCS (predictive and reactive control systems) theory [51,52,56,57]. According to the latter, the brain employs a reactive control system with feedback-guided mechanisms for handling novelty. The cerebral areas involved are the anterior insula, temporo-parietal junction, anterior hippocampal formation, ventral striatum, amygdala, and r-IFG. In particular, the r-IFG is engaged in the appraisal of stimuli, which is related to prediction and performance errors, novelty, and incongruity and induces the transition from model-guided feedforward control to momentary feedback-guided control [57]. This interpretation of the r-IFG’s role is consistent with its potential role in the inhibition of fast thinking.

Given the current debate and still uncertain results about the cerebral structures which underlie the two modalities of reasoning, in the following, we will describe a systematic meta-analysis about fMRI studies aimed at investigating the dichotomy of the dual process of thinking. Through the application of the activation likelihood estimation (ALE) algorithm, we aim to determine the pattern of activation associated with fast and slow thinking.

## 2. Materials and Methods

### 2.1. Literature Search and Selection

We conducted a systematic and comprehensive literature search to select relevant neuroimaging studies published up to 15 June 2022 using the databases PubMed (https://pubmed.ncbi.nlm.nih.gov/) and Web of Science (https://webofknowledge.com). The selected keywords were combined using the Boolean operators AND and OR. The PubMed search input was (“deductive reasoning” OR “inductive reasoning” OR “Wason selection task” OR “conditional rule*” OR “probabilistic reasoning” OR “normative reasoning” OR “belief bias” OR “heuristic*” OR “base rate neglect” OR “conjunction fallacy” OR “dual-process theory” OR “dual process theory” OR “fast thinking” OR “slow thinking” OR “cognitive reflection test” OR “analytical thinking” OR “intuitive thinking” OR “associative thinking” OR “deliberative thinking” OR “System 1” OR “System 2” OR “contextual effect” OR ”content effect”) AND (“fMRI” OR “PET” OR “neuroimaging”).

Additional studies were searched from the references of all identified publications. Eligibility was determined through a two-step procedure performed by three of the authors (G.Ga., G.Gr., and F.G.). First, the titles and abstracts of all identified articles were screened. In the second step, the full texts of the studies, according to predefined eligibility criteria, were independently examined (consistency was high: the number of papers with disagreement was <5%), and agreement was reached after a discussion (consistency among the authors was high: the number of papers with disagreement was <5%). Our study was conducted following the preferred reporting items for systematic reviews and meta-analyses (PRISMA) guidelines (Appendix A) [58].

The studies were included for the quantitative analyses if they met the following criteria: (1) whole-brain analysis (we excluded studies in which only results from ROI analyses were reported); (2) availability of coordinates of activation foci clearly provided either in Montreal Neurological Institute (MNI) or Talairach reference space; (3) clear interpretation of results in terms of fast and slow thinking reported in the paper. Typical examples are those of task contrasts: (i) normatively correct answers with belief-based responses; (ii) abstract, rule-based, and reflective strategies vs. heuristic, superficial strategies; (iii) distinction between slow and fast thinking obtained by means of fitting data to respective models of response. Studies conducted using relational and/or spatial reasoning tasks were excluded. Other exclusion criteria were different neuroimaging data analyses or procedures and samples including children or adolescents.

The MRI quality of the included studies was assessed based on a set of guidelines for the standardized reporting of MRI studies (Appendix A) [59,60].

### 2.2. Activation Likelihood Estimation (ALE)

The analysis of all data was carried out using the activation likelihood estimation (ALE) meta-analysis algorithm, which was implemented in GingerALE 2.3.6 software (www.brainmap.org/ale). The ALE algorithm, as described in previous methodological papers [61,62,63,64], is a coordinate-based meta-analysis that uses peak coordinates from functional studies as input. The procedure of ALE meta-analysis is summarized here, as it has been well documented in previous papers [62,65]. The ALE algorithm evaluates the convergence of activation foci from various neuroimaging studies, modeled as probability distributions, against null distributions of random spatial associations among studies while controlling for sample size. The non-additive algorithm [64] was used to minimize within-experiment effects. Inference was performed at the cluster level, which provides a better balance between sensitivity and specificity [62] compared with other methods. The cluster-forming threshold was set to *p* < 0.005, and the size of the resulting supra-threshold clusters was compared (with a threshold of *p* < 0.05) to a null distribution determined by 5000 permutations of the data. The studies selected are detailed in Table 1.

The neuroanatomical coordinates reported in Talairach space [79] were transformed into MNI space for all analyses. Whole-brain maps of the thresholded ALE images were visualized in Mango V.4.0.1 (http://rii.uthscsa.edu/mango/), an anatomical image overlay program, superimposed onto a standardized anatomical template.

## 3. Results

The PRISMA flow chart of article selection is illustrated in Figure 1. Our search yielded 69 potentially eligible studies. After full assessment of the papers, we found an inadequate number of studies that separately presented the activation of each thinking process compared with a baseline. Finally, 12 studies (for a total of 15 contrasts) from 2003 to 2020 presented contrasts of slow vs. fast thinking (or contrasts of fast vs. slow thinking) and allowed for the identification of the neural correlates associated with dual-process theory-based tasks in general, without differentiating between the two forms of thinking. From these studies, a cumulative number of 320 healthy subjects and 166 foci resulted. The main characteristics of the studies included in the analysis are reported in Table 1.

The ALE meta-analysis of the studies included (Figure 2, Table 2 and Appendix A) identified the largest-sized cluster (2376 mm^3^) to be centered in the medial frontal gyrus (mFG) and the superior frontal gyrus (sFG—pre-SMA) extending in the dorsal anterior cingulate cortex (dACC), followed by a cluster (1464 mm^3^) located in the left insula (Ins) including the left inferior frontal gyrus (l-IFG).

## 4. Discussion

The investigation of the neural basis of the dual-process theory of thought began with the pioneering studies by Goel and collaborators [41,80,81]. In the following twenty years, there has been an ever-growing interest in this issue from the neuroscience viewpoint. Studies based on fMRI have attempted to clarify some aspects, with all the same limitations of applying brain imaging techniques to tasks with a long time course as the ones employed within the cognitive literature. The general overview is a fragmented picture that, on the one hand, confirms the role of prefrontal structures in the thought process, but, on the other hand, failed to find a consensus about specific areas associated with the two thinking systems.

To the best of our knowledge, this is the first attempt to perform a meta-analysis with the ALE algorithm aimed at determining the brain areas involved in dual-process theory-based tasks. The algorithm converged on the medial frontal cortex, superior frontal cortex (two areas often considered crucial to performing inhibitory function [82]), anterior cingulate cortex (crucial to detecting and processing conflicting situations due to external stimuli; see [83,84]), insula (alertness and salience processes [85,86,87,88]), and left inferior frontal gyrus (an area that plays a role in inhibitory processes, among other functions [89,90,91]). As expected, we observed the activation of prefrontal regions associated with thought and cognitive control that are recurrently cited in the literature about the neural basis of dual-process theory (for example, the anterior cingulate cortex and its role in conflict detection; see [31]). Importantly, the identification of some of these specific regions may help to discriminate among different interpretations of the neural basis of the dual-process theory of thought.

As described in the introduction, the r-IFG’s role is supported by several studies [53,54,55], in line with the PARCS theory-based interpretation of the dual-process distinction [52,92]. However, the ALE algorithm did not converge on the r-IFG; this may be explained by the inability of the eligible studies to explicitly differentiate between fast and slow thinking. Instead, our results highlighted the role of the l-IFG in dual-process tasks. According to the PARCS theory, the l-IFG could integrate inconsistent information with internal representations employed by fast thinking [52,57]. Also, the role of the l-IFG is coherent with a recent meta-analysis, where Pan et al. [93] reported that participants exhibiting traits of impulsivity showed morphometric differences in the left inferior frontal gyrus. Behaviorally, fast-thinking responses are mostly produced by cognitively impulsive individuals (measured by means of the CRT [8]; see also Baron et al. [94]) as defined by the psychometric approach in the dual-process framework. Indeed, different thinking dispositions based on fast and slow thinking are measured in terms of resistance to impulsive answers in an intuitive but incorrect way. It is important to note that although the relationship between impulsivity traits and cognitive style is somewhat controversial [95], there is evidence of a relationship among impulsivity, the CRT, and inhibitory processes [96].

As pertaining to DMN involvement in the dual-process theory of thought [25,49,50], we only observed a partial overlapping between the areas involved in the DMN and the areas found by the ALE algorithm application. Even in this case, this does not necessarily represent evidence against the DMN hypothesis, given that our meta-analysis was able to identify the cerebral areas associated with dual-process theory-based tasks without differentiating between fast and slow thinking. Specifically, the DMN hypothesis regards the neural basis of fast thinking and not dual-process theory in general, so the fact that our algorithm did not converge on the whole DMN circuit is compatible with this idea.

Overall, the areas identified in this study are generally associated with the cognitive control circuit [46,47,97,98]. Keeping in mind the fundamental differences between dual-process research and inhibitory control (high-level tasks with response times lasting up to 15–30 s for the former vs. low-level tasks with a time course of milliseconds for the latter), these results are not surprising, given the similarities (i.e., conflict detection, the role of inhibition) in the theorization of these two fields [36,99,100,101]. However, the relationship between the measures and tasks commonly employed in the cognitive literature and the perceptual and attentional tasks utilized in inhibitory control research is far less investigated. Regarding this topic, a recent paper by Dorigoni et al. [102] observed that the connection between the dual-process theory of thought and attention processes is difficult to evidence given the complexity of measuring cognitive reflection (e.g., issues such as determining if it is a single construct; see [103,104]) and the presence of confounding variables (e.g., working memory, intelligence) that may mediate the relationship between thinking processes and tasks related to inhibitory control. More broadly, our results confirm the evident relationship between the tasks employed within the framework of the dual-process theory of thought and the literature on basic attentional processes, particularly cognitive control. This is, of course, not surprising, considering that Daniel Kahneman himself, in the late 1960s, delved into attentional processes, especially the interplay between attention and effort [24]. These reflections played a significant role in prompting Kahneman, along with their colleague, Amos Tversky, to initiate a research program on heuristics and biases. Thus, future research could benefit from a more intense dialogue between these seemingly distinct fields. This could include new research lines aimed at determining the neural underpinnings of the common ground between thought and attentional processes within cognitive neuroscience.

Despite the limited number of eligible studies in the literature, the total number of contrasts considered in this meta-analysis is 15, a quantity that guarantees sufficient statistical power [59,60,61,62]. However, the papers amenable to ALE analysis did not allow for contrasting fast-thinking- and slow-thinking-based decisions. This issue may be ascribed to methodological limitations that intrinsically characterize literature dual-process tasks. Indeed, fMRI features (temporal resolution limited by hemodynamic response time; need to make repeated measures over time) are generally incompatible with standard approaches in cognitive psychology, where response tasks may range in length up to 30 s. Another point is that reasoning tasks are generally not suitable to be administered over tens or hundreds of trials (as opposed to classic cognitive control tasks).

The current results do not allow for discrimination among serial, parallel, and hybrid frameworks [6]. The observed activation of areas associated with conflict detection (such as the ACC) is not discriminative, because conflict detection is a shared feature among the variety of specific dual-process models [30,31,33,34,35]. Indeed, the possibility of discriminating among models in light of neural activation requires (i) understanding which areas are active when the cognitive system is employing fast and slow thinking, respectively, and (ii) determining the time course of such activation instances with a high temporal resolution. Noteworthily, dual-process theory is a psychological construct, and it could be misleading to expect a one-to-one relationship between each system and a specific cerebral area [24]. The investigation of the neural basis of thinking should consider that we are dealing with two distinct planes: psychological theories on one hand and neuroscientific explanation on the other. As a matter of fact, despite the popularity of the dual-process theory of thought, evolutionary psychologists challenge certain aspects of this theory, such as its being overly simplistic or the existence of a domain-general processing mechanism (slow thinking) (see [2] for see for an up-to-date review of this debate). So, the complexity of thought processes and human decision making makes it necessary to approach any evidence on the neural correlates of related psychological processes with great caution.

Also, we must point out that the aim of a meta-analysis consists in answering specific research questions by combining results from multiple studies, with advantages such as enhanced statistical power, heightened precision, and comprehensive integration of findings. However, meta-analytic results should take into account the several issues by which they can be affected, such as study quality, heterogeneity, selection bias, and data availability [2]. In our case, the intrinsic limitations of the literature of the neural correlates of thinking surely affect the scope of the meta-analytic results. Future research should overcome this limitation, increasing the quality of these studies. For example, a first step could be promoting fMRI research that also includes conditions where specific systems’ activation is contrasted with a common baseline in order to apply an ALE meta-analysis able to differentiate between fast and slow thinking.

Concluding, this study represents further evidence of the strong connection between cognitive control processes and the dual-process theory of thought. We hope that this contribution may represent a first step in the direction of integrating the fMRI methodology employed in the cognitive control literature with the thinking and reasoning area of investigation. In this regard, future research will focus on similarities and differences between these two distinct fields in view of a better understanding of our thought processes.

## Figures and Tables

**Figure 1 brainsci-14-00101-f001:**
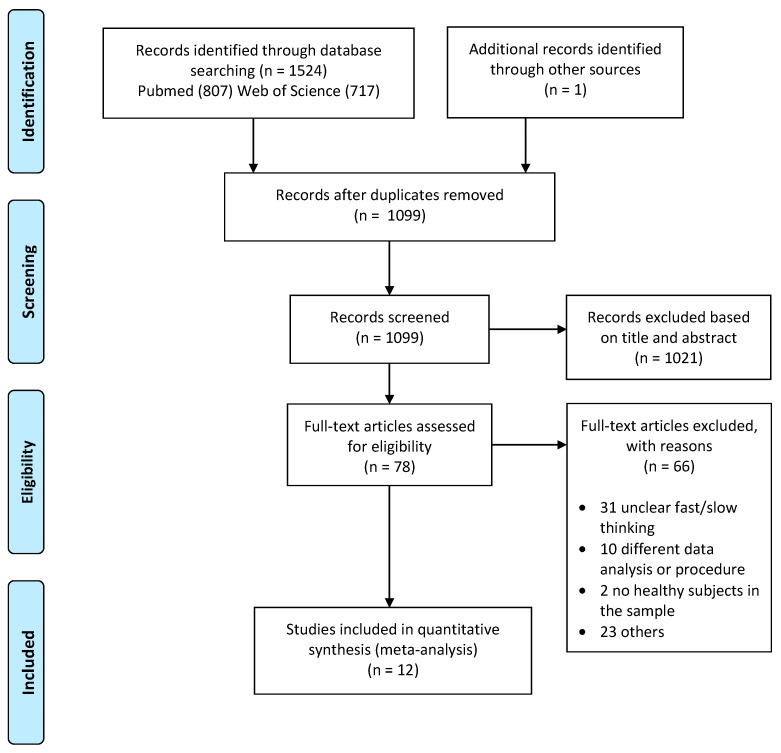
PRISMA flow chart.

**Figure 2 brainsci-14-00101-f002:**
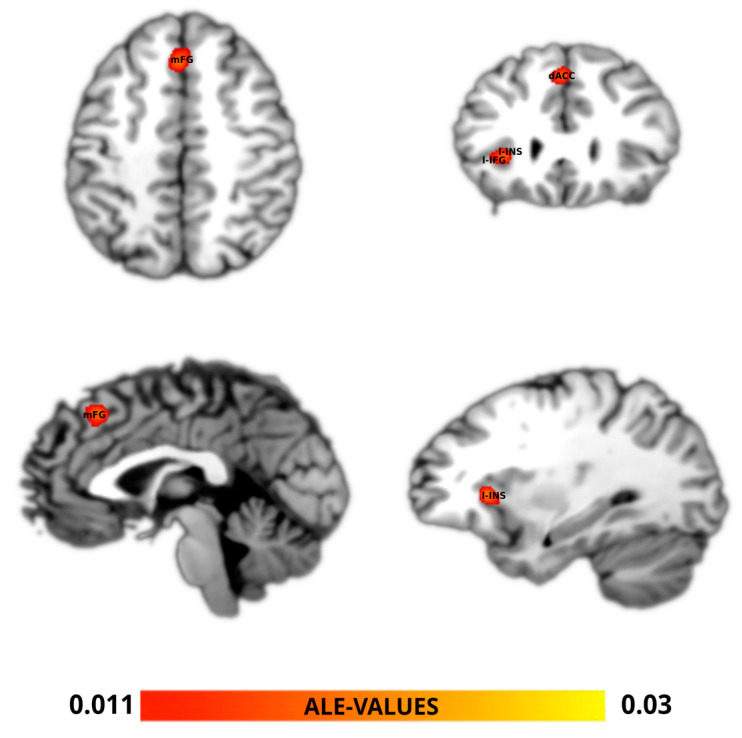
ALE meta-analysis map for the dual process of our data selection. The algorithm converged for the dual process (in yellow–red) on the medial frontal gyrus (mFG) and the superior frontal gyrus (sFG—pre-SMA), extending in the dorsal anterior cingulate cortex (dACC), and on the left insula (Ins) and inferior frontal gyrus (l-IFG). *p* < 0.05 cluster-level corrected inference using *p* < 0.005 uncorrected at voxel level as the cluster-forming threshold.

**Table 1 brainsci-14-00101-t001:** Characteristics of studies included in the analysis.

	Study	Year	Sample			Task	fMRI	
			Number	Age (Mean, SD or Range)	Gender (f/m)		Contrast	Number of Foci
1	Goel et al. [66]	2003	14	30.8 ± 4.3	7/7	Deductive reasoning task. A total of 120 syllogisms (15 different forms) and 40 baseline items organized in a nested 2 × 2 design (Belief × Task). The belief factor consisted of two levels: belief-laden (80 syllogisms and 40 baseline items) and belief-neutral (40 syllogisms and 20 baseline items) items. In the task factor, the first level (a reasoning condition) involved stimuli that constituted arguments (120 trials, as in the examples above and below). Half of these were valid, while the other half were not valid. The second-level (baseline condition) trials were generated by taking these arguments and switching around the third sentence, such that the three sentences did not constitute arguments.	Slow vs. Fast Fast vs. Slow	1/1
2	Canessa et al. [67]	2005	12	23.5 (21–26)	7/5	Deductive reasoning task. Two versions of the Wason selection task: the first version described an arbitrary relation between two actions (descriptive (DES): “If someone does …, then he does …”), whereas the other described an exchange of goods between two persons (social exchange (SE): “If you give me …, then I give you …”).	Fast vs. Slow	9
3	Beierholm et al. [68]	2011	23	n.r.	n.r.	Novel economic task, where three doors were visually presented. Participants were instructed to choose the order of the doors. After 6–8 s, the location of the money was revealed behind one of the doors, and subjects were rewarded according to the following: 0.50 USD if the money was behind their first choice, 0 USD if it was behind their second choice, and −0.50 USD if the money was behind the third choice. They were explicitly instructed to ignore anything they learned about the distribution of money and that the sequence of locations for the money was random. Behavioral data were employed to fit two models aimed at quantifying subjective valuation and updating signals corresponding to fast and slow thinking.	Fast vs. Slow	40
4	Liu et al. [69]	2012	14	21.8 (17–25)	6/8	Deductive reasoning task. Twenty-eight conditional reasoning statements (based on the Wason selection task).	Slow vs. Fast	9
5	Liang et al. [70]	2014	15	23.6 ± 3.1	7/8	Inductive reasoning task. One hundred twenty trials of a categorical induction task (modeled on stimuli from Osherson et al., 1990 [71]). Each trial was composed of pairs of arguments, and participants were instructed to indicate which one of the two arguments was stronger. Stimuli were divided into two conditions (explicit quantification vs. implicit quantification). Subjects’ responses to each trial were used to further divide the stimuli into fallacy or non-fallacy response trials.	Fast vs. Slow	6
6	Liang et al. [72]	2014	23	24.1 ± 3.7	11/12	Inductive reasoning task. Thirty number-series induction tasks, thirty letter-series induction tasks, twenty-four number judgment baseline tasks and twenty-four letter judgment baseline tasks were organized into a 2 × 2 factorial design (Content × Task). Content factor: number-related vs. letter-related content. Task factor: series completion task vs. baseline condition.	Fast vs. Slow	17
7	Luo et al. [73]	2014	16	23 (20–28)	8/8	Deductive reasoning task. One hundred twenty items (encompassing four different conditional reasoning forms) for the condition. Participants were required to draw a conclusion based on the premises. This study was organized into a 2 × 2 design (Type of Problem × Logical Training). Type-of-problem factor: conflict problems (in which the logical conclusion is inconsistent with one’s beliefs) and non-conflict problems (in which the logical conclusion is consistent with one’s beliefs). Logical training factor: naive participants vs. post logic training.	Slow vs. Fast	5
8	von Helversen et al. [74]	2014	23	20.13 ± 2.67	17/6	Categorial induction task. Participants were required to make fictitious quantitative judgements on 9 items (3 for each scenario) using a scale with 100 possible values. Each item was described by six binary cues and a criterion value. The task was based on learning to estimate the correct criterion value of items given the item’s cue values. Participants were instructed to use either a similarity-based exemplar strategy or a rule-based strategy. The actual use of the two strategies was determined by means of a computational model.	Slow vs. Fast Fast vs. Slow	7/9
9	Durning et al. [75]	2015	10 *	29.6 ± 2	3/7	Medical diagnosis task. Participants were presented with medical scenarios, and they were required to answer “what is the most likely diagnosis?” by choosing among 5 options. Participants were then given seven seconds to choose an answer option using finger response items, which would be expected to require both analytical and non-analytical reasoning. The final phase (“reflection” phase) was then entered; in this phase, participants were instructed to reflect on how they had arrived at the diagnosis, which primarily required (or accentuated) analytical reasoning.	Fast vs. Slow	17
10	Megìas et al. [76]	2015	56	22.24 ± 2.7	39/17	Novel risky driving evaluation task. Participants performed an urgent task (to brake or not in a given traffic situation) and an evaluative task (to evaluate whether the traffic situation entailed risk or not) during the experiment. Each task comprised 140 trials (70 risky situations and 70 non-risky situations).	Fast vs. Slow	21
11	Vartanian et al. [49]	2018	44	35.5 ± 11.3	13/31	Probabilistic reasoning task. Forty-eight base rate problems (24 conflict, 24 non-conflict) selected from Cheyne, et al.’s (2014) [77] item pool.	Slow vs. Fast	7
12	van den Berg et al. [78]	2020	16	51 (46–57)	4/12	Novel medical diagnosis task. Participants were required to diagnose 26 neurological cases. Each case had both fast-thinking (prototypical information) and slow-thinking (ambiguous information) versions to elicit the different types of reasoning.	Slow vs. Fast Fast vs. Slow	8/9

* Only data on novices were included.

**Table 2 brainsci-14-00101-t002:** Results from ALE meta-analysis. Foci are reported in MNI coordinates. BA = Brodmann area.

Cluster	x	y	z	*p*	Z	Label (Nearest Gray Matter within 5 mm)
1	−2	26	44	5.20 × 10^−7^	4.884085	Left cerebrum. Frontal lobe. Medial frontal gyrus. Gray matter. Brodmann area 8
	12	30	52	4.88 × 10^−4^	3.2970774	Right cerebrum. Frontal lobe. Superior frontal gyrus. Gray matter. Brodmann area 6
	8	24	34	0.0013236	3.0059886	Right cerebrum. Frontal lobe. Cingulate gyrus. Gray matter. Brodmann area 32
2	−30	24	4	1.68 × 10^−6^	4.6475234	Left cerebrum. Sub-lobar. Insula. Gray matter. Brodmann area 13
	−34	30	−2	9.45 × 10^−4^	3.1068544	Left cerebrum. Frontal lobe. Inferior frontal gyrus. Gray matter. Brodmann area 45

## Data Availability

The data analyzed in this meta-analysis are available in the studies listed in Table 1, and further inquiries can be directed to the corresponding author. The meta-analysis was not preregistered.

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
