# Peer review of "Dual-Process Theory of Thought and Inhibitory Control: An ALE Meta-Analysis"

_brainsci, 2024, doi:10.3390/brainsci14010101_

Round 1

Reviewer 1 Report

Comments and Suggestions for Authors

I like the links the authors draw between dual process and cognitive control literature.

 It is regretful that this analysis cannot get really to the point of the differentiation between the neural correlates of fast and slow thinking. It only identified the cerebral areas associated with dual-process theory-based tasks without differentiating between fast and slow thinking. This should be explicitly explained in the abstract and results section. In those places, it is confusing to understand what the results represent.

The inability to differentiate between fast and slow thinking is offered as an explanation of why the DMN was not clearly found in the results. This may equally-well explain why rIFG was not found. By contrast, lIFG may be implicated in “reappraisal” in attempts to integrate inconsistent information with internal representations employed by fast thinking [52, 57].

Please correct small grammatical errors in these sentences:

p. 3 line 118: Focusing on the latter, most studies have found a patterns of activation in prefrontal cortex, particularly in regions involved in cognitive control processes [36, 41-43].

p. 12 line 270: It is important to note that the albeit the relationship between impulsivity traits and

cognitive style is somewhat controversial [91], there are evidence of a relationship between impulsivity, CRT and inhibitory processes [92].

Author Response

I like the links the authors draw between dual process and cognitive control literature.

Point 1: It is regretful that this analysis cannot get really to the point of the differentiation between the neural correlates of fast and slow thinking. It only identified the cerebral areas associated with dual-process theory-based tasks without differentiating between fast and slow thinking. This should be explicitly explained in the abstract and results section. In those places, it is confusing to understand what the results represent.

Response: We have provided a more detailed specification of these aspects in the Abstract and in the Results sections.

Point 2: The inability to differentiate between fast and slow thinking is offered as an explanation of why the DMN was not clearly found in the results. This may equally-well explain why rIFG was not found. By contrast, lIFG may be implicated in “reappraisal” in attempts to integrate inconsistent information with internal representations employed by fast thinking [52, 57].

Response: We have added these points in the Discussion.

Point 3: Please correct small grammatical errors in these sentences:

- p. 3 line 118: Focusing on the latter, most studies have found a patterns of activation in prefrontal cortex, particularly in regions involved in cognitive control processes [36, 41-43].

- p. 12 line 270: It is important to note that the albeit the relationship between impulsivity traits and cognitive style is somewhat controversial [91], there are evidence of a relationship between impulsivity, CRT and inhibitory processes [92].

Response: These grammatical errors have been corrected.

Reviewer 2 Report

Comments and Suggestions for Authors

This article utilized ALE (Activation Likelihood Estimation) meta-analysis to examine the Dual Processes Theory of thought and inhibitory control. The layout is logical. The overall reception is positive. Here are some suggestions:

1. In the literature search, have you considered using other databases for literature search to ensure systematic and comprehensive coverage? Is it possible that some relevant studies were overlooked?

2. In line 177, it is mentioned that a two-step procedure was conducted by three authors. Is there consistency among the authors in determining eligibility? Are there any disagreements that need further discussion?

3. It is suggested to provide a more accurate description of the criteria for including articles and recommend adding an evaluation of the quality of included articles.

4. It is recommended to supplement the PRISMA checklist regarding the reporting of study characteristics.

5. Table 1 reports the main features of the studies included in the analysis, are there any other relevant features (such as study design, participant information, etc.) not mentioned in the text?

6. Can more detailed information be provided in Figure 2 and Table 2, such as specific brain region activation coordinates for each cluster and related studies?

Author Response

We thank the Reviewers for the very constructive comments. Below we outline how we have handled them.

Response to Reviewer 2 Comments

This article utilized ALE (Activation Likelihood Estimation) meta-analysis to examine the Dual Processes Theory of thought and inhibitory control. The layout is logical. The overall reception is positive. Here are some suggestions:

Point 1. In the literature search, have you considered using other databases for literature search to ensure systematic and comprehensive coverage? Is it possible that some relevant studies were overlooked?

Response: We agree with the Reviewer that consulting more databases is preferable to ensure a comprehensive coverage. Therefore, we added a literature search by the Web of Science, a database commonly used for ALE meta-analysis. In the present version of the manuscript the subsection ‘Literature search and selection’ of the Materials and Methods section and the PRISMA flow chart of study selection were modified accordingly. Eight additional articles were assessed for eligibility with respect to the previous version. However, all these new studies were excluded. Therefore, we confirmed that 12 studies are eligible for the quantitative synthesis.

Point 2. In line 177, it is mentioned that a two-step procedure was conducted by three authors. Is there consistency among the authors in determining eligibility? Are there any disagreements that need further discussion?

Response: The consistency among the authors was high, the number of paper with a disagreement was < 5%. Agreement was reached after discussion among authors. This information was added in the present version of the manuscript.

Point 3. It is suggested to provide a more accurate description of the criteria for including articles and recommend adding an evaluation of the quality of included articles.

Response: According to the referee’s suggestion, we added more details on inclusion/exclusion criteria in the Methods. Moreover, we performed an evaluation of the quality of included studies. Namely, the MRI procedures quality was assessed following a set of guidelines for the standardized reporting of MRI studies (Poldrack, R. A. et al. Guidelines for reporting an fMRI study. Neuroimage 40, 409–414, 2008). The table is provided as Supplementary materials (Table S1).

Point 4. It is recommended to supplement the PRISMA checklist regarding the reporting of study characteristics.

Response: In the present version of the manuscript we added the PRISMA checklist as supplementary material.

Point 5. Table 1 reports the main features of the studies included in the analysis, are there any other relevant features (such as study design, participant information, etc.) not mentioned in the text?

Response: Furher details are given in Supplementary Table S1 and S2.

Point 6. Can more detailed information be provided in Figure 2 and Table 2, such as specific brain region activation coordinates for each cluster and related studies?

Response: We thank the reviewer for this point. We added one supplementary table (Table S2) to detail the missing information.

Reviewer 3 Report

Comments and Suggestions for Authors

Following my examination of the paper Dual processes theory of thought and inhibitory control: an ALE meta-analysis, I recommend Minor Revision. The research presents a well-organized and methodologically sound meta-analysis that effectively synthesizes a variety of studies in order to investigate the brain foundations of the dual-processing theory of the mind. The application of Activation Likelihood Estimation (ALE) improves the scientific rigor of the work.

However, there are areas where clarity and depth could be improved. Specifically, the discussion could benefit from a more detailed exploration of the implications of these findings for the broader field of cognitive neuroscience and psychology. Additionally, a more thorough consideration of the limitations of the study and potential avenues for future research would strengthen the paper. These revisions are relatively minor but would significantly enhance the paper’s impact and readability.

Major comments

1. Authors should make it clear that we do not expect a one-to-one relationship between a System and a neurological correlate because neither System 1 nor System 2 has a single location in the brain [1]. It’s important to remember that the dual-process theory is a psychological construct, not a direct neural explanation. Therefore, there’s a distinct difference between psychological theories (mind) and neuroscientific findings (brain). Essentially, the mind is understood as a manifestation of brain activity [2].

2. Authors ought to exhibit greater nuance in their treatment of meta-analysis evidence. For example, meta-analyses generally support the dual-processing theory of the mind, finding evidence for the existence of two distinct cognitive systems and the idea that they can influence behavior and decision making in different ways. In contrast, certain researchers have contended that the theory lacks empirical backing and that the differentiation between System 1 and System 2 thinking is not distinctly outlined or universally applicable. They claim that the theory is based on anecdotal evidence. This criticism should not be dismissed in the face of supportive meta-analyses because meta-analyses alone cannot resolve an issue. A meta-analysis is a statistical method used to merge findings from several studies, aiming to derive a conclusion regarding a particular research question. Meta-analysis has the advantages of increased statistical power, improved precision, and finding integration. However, heterogeneity, study quality, selection bias, data availability, and model selection are all issues. Meta-analysis has grown in popularity in recent years. However, it is overused. For instance, textbooks improperly utilize their findings to resolve a controversy. See a discussion in [3].

Minor comments

1. Meta-analysis: see the title’s typo.

2. Line 27: Plato’s allegory contrasts reason and emotion. This is distinct from slow versus fast thinking. This is significant because, in Plato’s world, reasoning functions well unless it is disrupted by emotion. The contemporary contrast between slow fast thinking alludes to issues with the mind's machinery.

[1] Kahneman, D. Thinking, Fast and Slow; Farrar, Straus and Giroux: New York, NY, USA, 2011

[2] Kalat, J.W. Biological Psychology, 13th ed.; Cengage Learning, Inc.: Boston, MA, USA, 2019

[3] Psych 2023, 5(4), 1057-1076; https://doi.org/10.3390/psych5040071

Author Response

We thank the Reviewers for the very constructive comments. Below we outline how we have handled them.

Response to Reviewer 3 Comments

Following my examination of the paper Dual processes theory of thought and inhibitory control: an ALE meta-analysis, I recommend Minor Revision. The research presents a well-organized and methodologically sound meta-analysis that effectively synthesizes a variety of studies in order to investigate the brain foundations of the dual-processing theory of the mind. The application of Activation Likelihood Estimation (ALE) improves the scientific rigor of the work.

However, there are areas where clarity and depth could be improved. Specifically, the discussion could benefit from a more detailed exploration of the implications of these findings for the broader field of cognitive neuroscience and psychology. Additionally, a more thorough consideration of the limitations of the study and potential avenues for future research would strengthen the paper. These revisions are relatively minor but would significantly enhance the paper’s impact and readability.

Response: We added several parts in the discussion with a more detailed exploration of our findings along with the limitation and future research.

Major comments

Point 1. Authors should make it clear that we do not expect a one-to-one relationship between a System and a neurological correlate because neither System 1 nor System 2 has a single location in the brain [1]. It’s important to remember that the dual-process theory is a psychological construct, not a direct neural explanation. Therefore, there’s a distinct difference between psychological theories (mind) and neuroscientific findings (brain). Essentially, the mind is understood as a manifestation of brain activity [2].

Response: we have pointed out this aspect in the Discussion.

Point 2. Authors ought to exhibit greater nuance in their treatment of meta-analysis evidence. For example, meta-analyses generally support the dual-processing theory of the mind, finding evidence for the existence of two distinct cognitive systems and the idea that they can influence behavior and decision making in different ways. In contrast, certain researchers have contended that the theory lacks empirical backing and that the differentiation between System 1 and System 2 thinking is not distinctly outlined or universally applicable. They claim that the theory is based on anecdotal evidence. This criticism should not be dismissed in the face of supportive meta-analyses because meta-analyses alone cannot resolve an issue. A meta-analysis is a statistical method used to merge findings from several studies, aiming to derive a conclusion regarding a particular research question. Meta-analysis has the advantages of increased statistical power, improved precision, and finding integration. However, heterogeneity, study quality, selection bias, data availability, and model selection are all issues. Meta-analysis has grown in popularity in recent years. However, it is overused. For instance, textbooks improperly utilize their findings to resolve a controversy. See a discussion in [3].

Response: we have stated more clearly this aspect in the Discussion.

Minor comments

  1. Meta-analysis: see the title’s typo.

Response: We have corrected the title’s typo.

  1. Line 27: Plato’s allegory contrasts reason and emotion. This is distinct from slow versus fast thinking. This is significant because, in Plato’s world, reasoning functions well unless it is disrupted by emotion. The contemporary contrast between slow fast thinking alludes to issues with the mind's machinery.

Response: We have removed this reference.
